# Increased Renal Medullary NOX-4 in Female but Not Male Mice during the Early Phase of Type 1 Diabetes: Potential Role of ROS in Upregulation of TGF-β1 and Fibronectin in Collecting Duct Cells

**DOI:** 10.3390/antiox12030729

**Published:** 2023-03-16

**Authors:** Felipe Casado-Barragán, Geraldine Lazcano-Páez, Paulina E. Larenas, Monserrat Aguirre-Delgadillo, Fernanda Olivares-Aravena, Daniela Witto-Oyarce, Camila Núñez-Allimant, Katherin Silva, Quynh My Nguyen, Pilar Cárdenas, Modar Kassan, Alexis A. Gonzalez

**Affiliations:** 1Institute of Chemisry, Pontificia Universidad Católica de Valparaíso, Valparaíso 2950, Chile; 2Skaggs School of Pharmacy and Pharmaceutical Sciences, University of California San Diego, La Jolla, CA 92093, USA; 3College of Dental Medicine, Lincoln Memorial University, Knoxville, TN 37917, USA

**Keywords:** diabetes, ROS, sex differences, mice, fibrotic factors

## Abstract

Chronic diabetes mellitus (DM) can lead to kidney damage associated with increased reactive oxygen species (ROS), proteinuria, and tubular damage. Altered protein expression levels of transforming growth factor-beta 1 (TGF-β1), fibronectin, and renal NADPH oxidase (NOX-4) are associated with the profibrotic phenotype in renal tubular cells. NOX-4 is one of the primary sources of ROS in the diabetic kidney and responsible for the induction of profibrotic factors in collecting duct (CD) cells. The renal medulla is predominantly composed of CDs; in DM, these CD cells are exposed to high glucose (HG) load. Currently there is no published literature describing the expression of these markers in the renal medulla in male and female mice during the early phase of DM, or the role of NOX-4-induced ROS. Our aim was to evaluate changes in transcripts and protein abundances of TGF-β1, fibronectin, and NOX-4 along with ROS levels in renal medullary tissues from male and female mice during a short period of streptozotocin (STZ)-induced type 1 DM and the effect of HG in cultured CD cells. CF-1 mice were injected with or without a single dose of STZ (200 mg/kg) and euthanized at day 6. STZ females showed higher expression of fibronectin and TGF-β1 when compared to control mice of either gender. Interestingly, STZ female mice showed a >30-fold increase on mRNA levels and a 3-fold increase in protein levels of kidney medullary NOX-4. Both male and female STZ mice showed increased intrarenal ROS. In primary cultures of inner medullary CD cells exposed to HG over 48 h, the expression of TGF-β1, fibronectin, and NOX-4 were augmented. M-1 CD cells exposed to HG showed increased ROS, fibronectin, and TGF-β1; this effect was prevented by NOX-4 inhibition. Our data suggest that at as early as 6 days of STZ-induced DM, the expression of profibrotic markers TGF-β1 and fibronectin increases in renal medullary CD cells. Antioxidants mechanisms in male and female in renal medullary tissues seems to be differentially regulated by the actions of NOX-4.

## 1. Introduction

The epidemic of diabetes mellitus (DM) is progressively worsening worldwide and is a major cause of kidney failure [1]. Although hyperglycemia is often the first symptom of DM, other hallmarks include hyperglycosuria, proteinuria, and other signs of renal damage, including deposition of extracellular matrix and glomerular damage along with tubular fibrosis [2]. These clinical manifestations are primarily observed in the advanced disease. In general, the diabetic animal models reported for the study of diabetic disease have been used to analyze the effects of chronic hyperglycemia and glycosuria on renal physiology and on the expression of kidney injury markers related to glomerular damage [3,4].

Among the injury markers observed in chronic DM in experimental animal models, transforming growth factor-beta 1 (TGF-β1), fibronectin, and connective tissue growth factor (CTGF) have been found to be responsible for the profibrotic effects in tubular cells; this enhances collagen deposition and the proliferation of fibroblasts [5,6]. Tubular markers of injury in diabetic disease are associated with proteinuria and albuminuria; however, the progression of tubule-interstitial disease, in addition to glomerular injury in diabetes, is also important and may provide insights into the pathogenesis of diabetic nephropathy beyond the glomerular injury.

Because the renal medullary tissues and the collecting ducts are more likely to be damaged when limited glucose transport in the proximal tubule has been exceeded, their resorptive function is especially affected by the pathological mechanisms activated by hyperglycemia and hyperglycosuria. The exposure to high glucose levels promotes an increase in cytokines levels such as TGF-β1, as well as extracellular matrix formation in the kidney [7]. Tubular and interstitial expression of profibrotic factors thus amplifies the development of fibrogenesis in the renal tubulointerstitium [8]. Advanced glycation also occurs in in the diabetic state, promoting increased reactive oxygen species (ROS) generation. Additionally, it has been reported that the mechanisms underlying hyperglycemia and hyperglycosuria-induced renal complications involve renal NADPH oxidase (NOX-4) [9]. This further increases ROS generation, causing activation of redox-sensitive pathways in DM [10]. In early DM, hyperglycemia and hyperglycosuria may affect redox status in kidney cells; this may favor cell signaling pathways that enhance extracellular matrix and damage in renal tubules [11,12]. Then, increases of ROS in renal epithelial cells may be the mechanism by which renal tubular injury occurs [13,14].

Most of these effects are observed in longstanding hyperglycemia (lasting weeks or longer) and all current data in the literature are primarily focused on male mice. Sex differences in glucose metabolism have been described in animal models of DM [15]. It has been observed that glucose metabolism can be affected by variations in hormone levels through the estrogen cycle. Evidence showed that insulin resistance can be observed in male STZ mice, but not in female STZ mice, after 6 weeks of STZ injection [15,16]. This indicates that glucose metabolism in the diabetic male mice is worse than that observed in female STZ mice, which thereby suggests that estrogen may play an important role in glucose metabolism in STZ-induced DM. We evaluated the physiological parameters and renal medullary expression of TGF-β1, fibronectin, and NOX-4 in male and female mice with streptozotocin (STZ)-induced diabetes mellitus over the course of 6 days following STZ treatment. We also tested the effect of 48 h of high glucose conditions on the expression of profibrotic markers in cultured inner medullary collecting duct cells (IMCD). To further avoid the hormonal and sex influence in cultured cells as potentially confounding variables, we performed additional experiments in M-1 collecting duct cell line to evaluate the role of NOX-4.

## 2. Materials and Methods

### 2.1. Animals and Samples

All methodologies were in accordance with the Animal Care and the Bioethical Committee of the Pontificia Universidad Católica de Valparaiso (code number BIOEPUCV-B 267-2019), under international guidelines and regulations for animal use. Male and female CF-1 mice (12-week-old) were placed under a 12 h light–dark cycle, a temperature of 21 °C, humidity of 50%, noise-free conditions, and food and water ad libitum. Male and female mice were divided randomly into two groups: control (saline injection, normoglycemic, *n* = 6) and streptozotocin (STZ)-induced (200 mg/kg, single i.p. injection, *n* = 6). STZ was injected after a 6 h fast [17]. A diabetic mouse was considered if three consecutive blood glucose readings exceeded 250 mg/dL. At the end of the treatment period, mice were euthanized by cervical dislocation after isoflurane inhalation. A graphical abstract including the protocols of animal and cell cultures is shown in Figure 1.

### 2.2. Blood Glucose Measurements

Blood glucose was directly measured using a ONETOUCH Ultra glucometer (LifeScan, catalog #ZJZ8158JT, Milpitas, CA, USA, reported result range 20–600 mg/dL) and compared with a regular glucometer Prodigy© (Charlotte, NC, USA) demonstrating no differences in plasma glucose measurements. A glucose overload test was performed once at day 5 after STZ treatment. The blood glucose was measured before and after 15, 30, 60, and 120 min of a glucose overload of 2 g/kg BW, i.p. Blood glucose was measured every 15 min.

### 2.3. Sodium Measurement from Plasma and Urine

Sodium was measured from plasma and urine samples collected from metabolic cages on day 6. Sodium was measured using a flame photometer (Instrumentation Laboratory 943, Ramsey, MN, USA).

### 2.4. Blood Creatinine

Blood was collected from the cardiac puncture and centrifuged at 10,000× *g* for 5 min at 4 °C to collect the plasma. Creatinine was measured using a Creatinine Analyzer 2 (Beckman Coulter, Inc., Fullerton, CA, USA).

### 2.5. Blood Pressure Measurements

The mice were trained (4 days) to the tail-cuff plethysmography protocol, eight to 12 consecutive pulse readings were recorded for each mouse in each set of measurements on day 0, 3, and 6. All data were recorded using a BP-2000 series II Blood Pressure Analysis System (Visitech Systems, Inc.; Apex, NC, USA).

### 2.6. Saline Challenge

On day 5, a saline challenge was performed to evaluate the effect of STZ on Na^+^ balance. Mice were injected (i.p.) with isotonic saline (10% of their body weight) and placed in metabolic cages for urine collection.

### 2.7. Assays for Reactive Oxygen Species in Medullary Tissues

Total ROS in renal homogenates and plasma were measured with the Oxiselect^™^ in vitro ROS/RNS assay kit (Cell Biolabs, Inc. San Diego, CA, USA), following the manufacture’s instruction.

### 2.8. Fibronectin, TGF-β1 and NOX-4 Transcripts Quantitation by Real Time qRT-PCR

The mRNA was extracted from renal medullas (1 mm × 1 mm square tissue) using a RNeasy Mini Kit (Qiagen, Valencia, CA, USA). The RNA was quantified using the nano-drop system. Quantitative real-time RT-PCR (qRT-PCR) was performed using the following primers (5′–3′): TGF-β1: TCGCTTTGTACAACAGCACC (Forward); ACTGCTTCCCGAATGTCTGA (Reverse), gene accession number NM_011577.2; Fibronectin: TCCCGGGCAGAAAGTACATT (Forward), TTCAGGGAGGTTGAGCTCTG (Reverse) gene accession number M18194.1; NOX-4: TGCTCATTTGGCTGTCCCTA (Forward), TGCAGTTGAGGTTCAGGACA (Reverse), gene accession number NM_015760.5. Results were presented as the fold change ratio between the levels of mRNA of the interest gene against β-actin (“housekeeping” gene, gene accession number NM_023063.4) and compared to control group (*n* = 4–6). Primers were obtained from IDT Company (https://www.idtdna.com, Newark, NJ, USA).

### 2.9. Immunoblotting Analyses

Inner sections of kidney poles were minced and homogenized in ice-cold RIPA buffer with protease inhibitors (0.15 M NaCl, 50 mM Tris-HCl pH 8.0, 1% NP-40, 0.5% deoxycholate). Total protein content was determined using the BCA Protein Assay Kit (Santa Cruz, CA, USA) according to manufacturer instructions. Protein samples (40 micrograms) were separated on a precast NuPAGE 10% Bis-Tris gel (Novex) at 200 v for 45 min. Blots were blocked at 25 °C and incubated with primary antibodies. Then, the membranes were incubated with the secondary antibodies (1:1000 dilutions) and analyzed by normalization against β-actin bands (used as a housekeeping gene). Fibronectin, TGF-β1, and NOX-4 protein levels were detected using polyclonal rabbit antibody from Santa Cruz, CA, used at 1:200 dilutions. Immunoblots are shown in each figure as representative images. Results are shown as the ratio of each band vs. β-actin (fold change of control). Analysis was conducted by using 4–6 animals per group and 3–5 independent experiments for Western blot analysis (Appendix A). Blots were incubated with chemiluminescent substrate ECL reagent (Perkin Elmer, Waltham, MA, USA) by direct exposure on films that were scanned and analyzed by ImageJ software.

### 2.10. Primary Cultures of Inner Medullary Collecting Duct (IMCD) Cells

IMCD cell cultures were prepared using inner medullary tissues (no glomeruli components). Inner medullary tissues were minced and digested in 10 mL of DMEM-Ham F-12, 20 mg of collagenase B, 7 mg of hyaluronidase, 80 mmol/L of urea, and 130 mmol/L of NaCl and incubated at 37 °C with agitation for 90 min. The pellet was washed in p culture medium without enzymes. The IMCD cell suspension was seeded in 3-mm petri dishes until 70% confluence (3–5 days) and incubated with normal (5 mM) or high (25 mM) glucose conditions. IMCD cells were obtained from female or male mice. Preliminary experiments demonstrated that IMCD cells from male and female showed same profile of expression in response to normal or high glucose (Appendix A). Since both IMCD cultures were under no control of sex hormones, we decided to perform the experiments in male IMCD. Blot analysis was conducted by using 4 replicates per group (Appendix A).

### 2.11. Immunofluorescence in Kidney Slides and IMCD Cells

For histological studies, a kidney pole was fixed with Bouin’s solution (picric acid, formaldehyde and acetic acid 4%) overnight and then subjected to inclusion into paraffin blocks. Kidney poles were cut in 3 μm sections using microtome (HM325, Thermo, Waltham, MA, USA). Kidney slides (3 μm) were fixed and stained with antibodies at 1:100 dilutions and detected with Alexa Fluor 594 conjugated to antirabbit IgG (Invitrogen, Life Science, Co., Waltham, MA, USA). The slides were mounted with ProLong^®^ Gold. Subconfluent IMCD cells were fixed in cold methanol for 20 min, blocked with PBS-Tween (0.1%) plus BSA (3%), stained with Fibronectin, TGF-β1 and NOX-4 at 1:100 dilutions, and detected with Alexa Fluor 488 (Invitrogen Life Science, Co.). Samples were counter-stained with 4,6-diamidino-2-phenylindole (Invitrogen). Omission of the specific primary antibody was used as a negative control. The images were obtained using a Nikon Eclipse-50i (Nikon Eclipse-50i, Minato City, Japan) and NIS-Elements BR version 4.0 from Nikon.

### 2.12. M-1 Cell Culture

M-1 cells (ATCC, VA) cell line possess the phenotype of collecting duct cells [18]. The M-1 cells were cultured as previously described [19]. Cells were harvested after 48 h of treatments with normal or high glucose. NOX-4 inhibitor GKT 137831 was used at 30 μM [20]. The pharmacological inhibitor was added 30 min before high glucose incubations. Controls were performed with vehicle (DMSO, 0.06% vol/vol).

### 2.13. Statistical Analyses

Results are shown as mean ± SEM. The statistical analyses were performed with GraphPad Prism Software Version 6 (GraphPad Software, Inc., La Jolla, CA, USA). Normal distribution of each parameter analyzed was tested by using Shapiro–Wilk. Two-way ANOVA was used to compare the mean differences between groups and divided on variables controls vs. STZ in male and female mice studies and post-test comparisons for two groups using non-paired (one-tailed) *t*-test.

## 3. Results

### 3.1. Physiological Parameters in Male and Female STZ Mice

As shown in Table 1, STZ mice had increased fasting blood glucose (FBG) and reduced body weight at day 6, in a phenotype consistent with diabetic disease. No changes were seen in hematocrit electrolyte balance and serum creatinine. Food and water intake and urine output were increased in both male and female STZ mice when compared to control mice.

To ensure consistent expression of type 1 diabetes mellitus in STZ-induced mice, we performed a tolerance glucose test at day 5. As shown in Figure 2A, the STZ mice were not able to recover control levels of FBG when compared to control animals. Since we previously demonstrated that STZ mice showed Na^+^ retention [21], control and STZ mice were injected with isotonic saline (i.p. 10% of body weight) and placed in metabolic cages for urine measurements. A slight but not significant trend toward Na+ retention was observed in male and female STZ mice (Figure 2B). Interestingly, proteinuria did not differ between STZ females and control females; however, this parameter was significantly increased in STZ males (Figure 2C). Renal medullary levels of ROS were significantly augmented in female mice when compared to males (Figure 2D).

### 3.2. Expression of TGF-β1, Fibronectin and NOX-4 in Renal Medullary Tissues from Male and Female Mice after 6 Days of Streptozotocin (STZ)-Induced Type 1 Diabetes

After 6 days of STZ administration, NOX-4 mRNA/β-actin mRNA ratio was augmented in female STZ mice when compared to control female mice (39.2 ± 5.1 vs. control 2.5 ± 0.7, *p* < 0.001); this increase was not observed in male STZ mice (Figure 3A). Fibronectin mRNA/β-actin mRNA ratios were augmented in female (8.7 ± 0.6 vs. control 2.8 ± 0.6 *p* < 0.01) and male STZ mice (5.3 ± 0.4 vs. control 1.6 ± 0.5 *p* < 0.05, Figure 3B). Similarly, TGF-β1 mRNA/β-actin mRNA ratios were significantly augmented in STZ females vs. control female mice (3.2 ± 0.2 vs. control 0.8 ± 0.1, *p* < 0.05), and male STZ mice (2.0 ± 0.2 vs. control 0.9 ± 0.1, *p* < 0.05, Figure 3C).

Next, we evaluated protein abundance of these three profibrotic markers. Similar to what was observed in mRNA levels, NOX-4 was increased in STZ females (2.7 ± 0.1 vs. 1.0 ± 0.1, *p* < 0.01) but not in male STZ mice (Figure 4A). We also found significant differences in the abundance of fibronectin protein in STZ females vs. controls (2.7 ± 0.7 vs. 0.9 ± 1, *p* < 0.05, Figure 4B) and male STZ mice (1.7 ± 0.4 vs. 0.8 ± 0.1, *p* < 0.05, Figure 4B). We found increased levels of TGF-β1 1 in both females (1.8 ± 0.2 vs. 0.9 ± 0.1, *p* < 0.05) and males (1.9 ± 0.1 vs. 0.9 ± 0.1, *p* < 0.05, Figure 4C). A representative blot is shown in Figure 4D.

By using immunofluorescence, we analyzed renal cortical and medullary staining using fibronectin, TGF-β1, and NOX-4 antibodies. High intensity staining was evident for NOX-4 in STZ female mice. In males, high-intensity staining of NOX-4 was particularly noted in the cortex and in some interstitial cells, as judged by their shape. On the other hand, staining against fibronectin and TGF-β1 occurred primarily in tubular cells (Figure 5).

### 3.3. Expression of TGF-β1, Fibronectin and NOX-4 in Primary Cultured Inner Medullary Collecting Duct Cells Exposed to Normal and High Glucose Conditions

We tested the effect of high glucose during 48 h of incubation with 25 mM of glucose in cultured IMCD. As shown in Figure 5, the mRNA/β-actin mRNA ratios were augmented for fibronectin (1.62 ± 0.89 vs. control 1.00 ± 0.12 *p* < 0.01, Figure 6A), TGF-β1 (2.11 ± 0.1 vs. control 1.06 ± 0.03, *p* < 0.05, Figure 6B), and NOX-4 (1.63 ± 0.34 vs. control 1.00 ± 0.08, *p* < 0.05). An appropriate control for osmolality was evaluated by using 25 mM of mannitol; it showed no differences on the expression of profibrotic genes analyzed (Appendix A).

Using the same protocol, we performed an analysis of protein extracts from IMCD lysates after incubations with high glucose. As observed in Figure 7, protein abundance was significantly augmented for fibronectin (1.54 ± 0.12 vs. control 1.00 ± 0.05, *p* < 0.05), TGF-β1 (2.25 ± 0.18 vs. control 1.00 ± 0.07, *p* < 0.05), and NOX-4 (1.75 ± 0.08 vs. control 1.00 ± 0.08, *p* < 0.05).

Next, we performed immunofluorescence studies on IMCD cells using fibronectin, TGF-β1, and NOX-4 antibodies. IMCD cells showed a basal expression of fibronectin that was augmented after 48 h of HG. The TGF-β1 antibody did not show specific immunofluorescence labeling in IMCD cells. NOX-4 labeling intensity was present both at baseline and after 48 h of HG, as observed in Figure 8.

### 3.4. Effect of NOX-4 Inhibition on the mRNA Levels of Fibronectin and TGF-β1 in M-1 Collecting Duct Cell Line Exposed to High Glucose Conditions

To further evaluate the role of NOX-4 in an independent cell culture not influenced by sex, we performed a new protocol by using 30 μM of NOX-4 inhibitor GKT 137831 that was added 48 h before high glucose incubations. Results showed that GKT 137831 partially impairs ROS production and induction of mRNA levels for fibronectin and TGF-β1; this indicates that NOX-4 activity mediates glucose induction of ROS (Figure 9).

## 4. Discussion

In this study we demonstrated that both male and female CF-1 mice with 6 days of STZ-induced hyperglycemia exhibited increased fasting blood glucose, reduction in body weight, increases in food and water intake, and urinary output. All these symptoms are consistent with the clinical manifestation of DM in humans. Interestingly, determinations of Na^+^ balance among the groups using a Na^+^ challenge demonstrated a positive Na^+^ balance on day 6 in male and female STZ mice; this is consistent with previous studies using this model [21]. Because of the impact that positive Na^+^ balance may have on systolic blood pressure, we performed tail cuff—systolic blood pressure measurements in all groups. The results indicated that STZ treatment at day 6 had no impact on blood pressure in female and male mice. Despite the evident increase in urine proteinuria and the transitory Na^+^ positive balance ratio observed in STZ males, no impact on systolic blood pressure was observed. This suggests that blood pressure is not affected during the early stages of type I DM. Despite this observation, a direct effect of diabetic disease on blood pressure has been demonstrated in women [22,23,24,25]. By measuring systolic blood pressure using tail-cuff method, Islam et al. found that systolic blood pressure was increased 30 days post STZ injection [26]. More recently, Chandramouli et al., using a protocol of 5 consecutive daily injections of STZ (55 mg/kg), demonstrated that female mice exhibit a heightened susceptibility to diastolic dysfunction and a lower extent of hyperglycemia than male mice [27]. All this evidence may be also related to the peripheral and renal factors resulting in altered volume handling, thereby leading to high blood pressure. This concept is supported by prior studies, which showed paralleled increased levels in intrarenal renin angiotensin system and profibrotic and proinflammatory factors in response to high blood pressure and diabetic models [28,29,30,31].

In the present study, we used a 6-day STZ-induced diabetic model that is consistent with a condition of no gross kidney damage observed in chronic conditions. Longer periods of treatment are associated with renal and tubular damage, including collagen deposition in the glomeruli and tubular structures in rat [32] and mice kidneys after 3 and 5 weeks post-STZ injection, respectively [33]. Even longer treatments using a single STZ dose have shown evidence of tubular necrosis in mice [34]. To examine the impact of STZ treatment on glomerular function, we evaluated plasma creatinine and proteinuria. No significant difference was found in plasma creatinine levels between STZ mice and control mice of either gender. However, we did find a significant increase in urinary proteinuria in STZ males that was not observed in STZ females. This finding may be related to both a mild or incipient glomerular damage due to filtration of low molecular weight proteins and to an impairment of protein reabsorption in proximal tubules and tubulointerstitial disease [35,36].

Long-standing and uncontrolled diabetes is associated with end-organ complications, including diabetic kidney disease. Sex may play a role in the risk of progression to these dreaded complications. For example, women with diabetes have higher mortality rates and higher prevalence of diabetic kidney disease [37,38,39,40]. In contrast, a recent review by Giandalia et al. noted that the risk of developing and worsening diabetic kidney disease is higher in men with DM, while women are at higher risk of glomerular damage. These sex-dependent differences appear to exist in both type 1 and type 2 diabetes mellitus and thereby have implications in the diagnosis and management of DM-induced renal disease. The sex and gender differences in diabetic disease remain to be elucidated; however, hormonal and genetic differences have an impact in the development of renal injury [41].

Advanced glycation products, which promote ROS formation similarly to NOX-4, are present in collecting ducts; there, they are involved in the activation of redox-sensitive pathways in DM [10]. In early DM, hyperglycemia and hyperglycosuria may affect redox status in kidney cells; this may predispose to certain cell signaling pathways such as stimulation of TGF-β1 and extracellular matrix deposition in renal tubules [11,12]. We found that STZ mice showed higher expression of fibronectin and TGF-β1 compared to control mice of both genders. The increased levels of mRNA in STZ mice were more pronounced in females than in males. This finding does not correlate with the significant increases in proteinuria observed in STZ males, which indicates that this effect may be related to glomerular alterations during the early phase and not to damage associated to medullary collecting ducts. On the other hand, STZ female mice showed a near 40-fold increase in mRNA levels and a 3-fold increase in protein levels of kidney medullary NOX-4. Although both male and female STZ mice showed increases in medullary ROS, the increase in female STZ was more evident (Figure 1D). All this evidence was further demonstrated by immunofluorescence in kidney slides showing differences in expression of fibronectin, TGF-β1, and NOX-4 in renal cortex and medullary tissues. The induction of NOX-4 was primarily observed in renal medulla. Similar results were seen for TGF-β1.

Tor further evaluate the role of NOX-4 in ROS production, we performed experiments on M-1 cells lines, as described before [20]. Treatment with NOX-4 inhibitor GKT 137831 in the presence of HG conditions partially impairs ROS production and induction of mRNA levels for fibronectin and TGF-β1. This indicates that glucose-mediated induction of ROS is partially due to NOX-4 activity (Figure 10).

It has been shown that augmented ROS causes the secretion of active TGF-β1 protein complex in the extracellular matrix. Activated TGF-is a paracrine pathway that causes further stimulation of NOX-4 production. TGF-β1 also stimulates the fibrotic process [42]. Additionally, increases in TGF-β1, fibronectin, and NOX-4 (IMCD) were observed after 48 h of high glucose treatment in primary cultures of renal inner medullary collecting duct cells (Figure 7).

By promoting auto-oxidation of glucose to form free radicals, hyperglycemia can lead to microvascular dysfunction. Augmented proteinuria was observed in females and males; however, this was more evident in males. This may reflect a different mechanism of progression in glomerular capillary damage that might be determined by sex. The inhibition of ROS formation may provide a therapeutic strategy to prevent hyperglycemia-related oxidative stress during the early phase of diabetes. Because the inhibition of NOX-4 does not entirely suppress the augmentation of ROS in M-1 cells, it is possible that other mechanisms, such as scavenge defense capacity, may be responsible for the changes observed in renal cells. Then, antioxidants may be beneficial to inhibiting the damaging effects of DM by exerting their effects through variety of mechanisms, such as by inhibiting the formation of ROS and scavenging free radicals. They also promote nitric oxide (NO) production, which may improve endothelial dysfunction in DM. Antioxidants may also decrease vascular NOX-4 activity [43].

## 5. Limitation of the Study

The maximal sustained hyperglycemia is observed at day 6; thus, the mechanisms involved in the adaptative responses during lower intratubular and plasmatic glucose concentrations cannot be evaluated since an extremely brief period makes it difficult to perform physiological assays. In addition, it is possible that glucose handling in other organs may be affected by variations in sex hormones (i.e., through the estrogen cycle). We were unable to demonstrate this effect; however, future studies are proposed to evaluate the influence of female or male hormones in ovariectomized and castrated mice. Indeed, it has been shown that ovariectomized STZ female mice treated with estradiol showed a reduction in hyperglycemia [44]. More importantly, we cannot rule out the fact that the toxic effects of STZ on pancreatic islet β cells may also cause the low-grade inflammation in multiple organs, such as skeletal muscle and liver, causing alteration in pathways related to insulin metabolism [45].

## 6. Conclusions

In conclusion, our STZ-induced type 1 diabetes mellitus models demonstrated that in, as few as 6 days under conditions of hyperglycemia and glucosuria, levels of profibrotic markers TGF-β1 and fibronectin become increased in renal medullary tubular cells. The mechanism of this change involves NOX-4-dependent ROS formation. Of special note, NOX-4 was induced to a greater extent in female mice compared to their counterparts, suggesting that the mechanisms of regulation in IMCD cells are influenced by sex. Although our study did not evaluate the influence of sexual hormones, necessary approaches should be utilized by scientists and clinicians to develop a better understanding of the role of sex hormones in the pathophysiology of diabetic kidney disease, as well as the role of antioxidant pharmacological therapies.

## Figures and Tables

**Figure 1 antioxidants-12-00729-f001:**
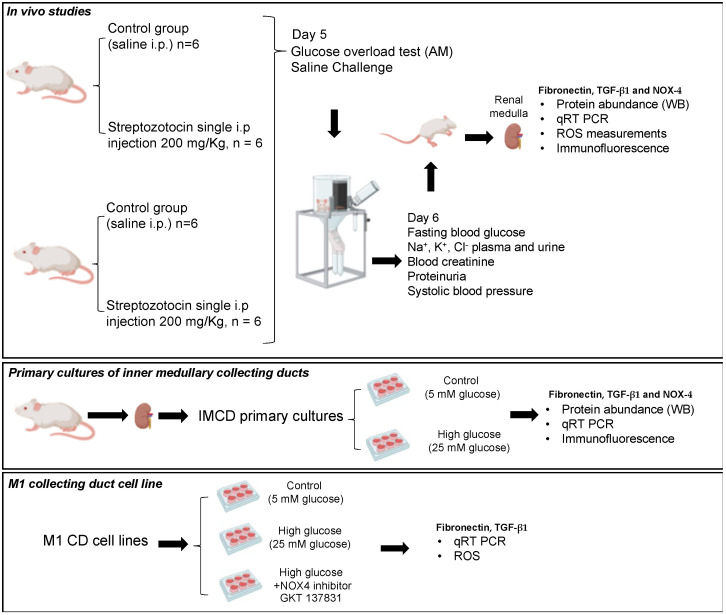
Graphical abstract showing all methods of the study. Three protocols were performed including studies in vivo using streptozotocin-induced type 1 diabetes, in vitro studies using primary cultures of kidney inner medullary collecting ducts treated with normal or high glucose for 48 h, and studies using M1 collecting duct cell line to test the effect of NOX4 inhibitor in conditions of high glucose on NOX4, fibronectin, and TGF-β1 expression.

**Figure 2 antioxidants-12-00729-f002:**
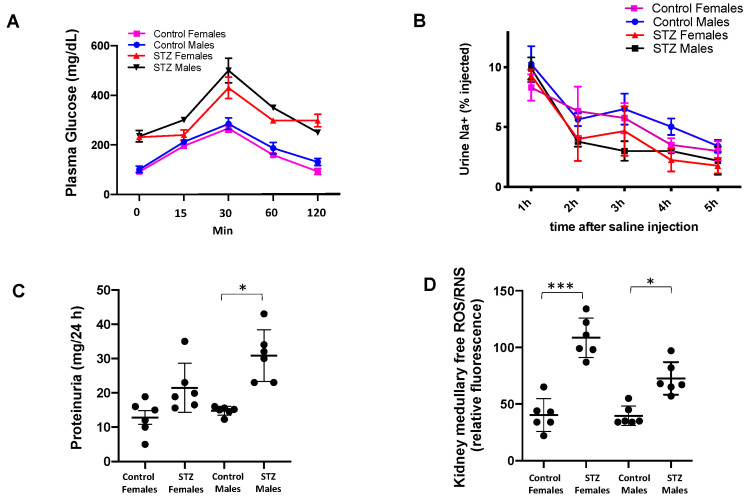
Plasma glucose, Na^+^ retention after saline challenge, proteinuria, and kidney ROS in male and female control and STZ mice. (**A**) Phenotype of type 1 diabetes mellitus was confirmed by a tolerance glucose test at day 5. STZ mice were not able to recover basal levels of FBG when compared to control animals. (**B**) Control and STZ mice were injected with isotonic saline (i.p. 10% of body weight) and in placed metabolic cages for Na^+^ urine measurements. STZ treatment caused Na^+^ retention in STZ males and females when compared to control group. (**C**) Proteinuria was significantly increased in STZ males but not in STZ female mice relative to their counterparts. (**D**) Kidney homogenates were used to measure free ROS/RNS levels; these were augmented in the female (*** *p* < 0.001) and male (* *p* < 0.05) STZ groups.

**Figure 3 antioxidants-12-00729-f003:**
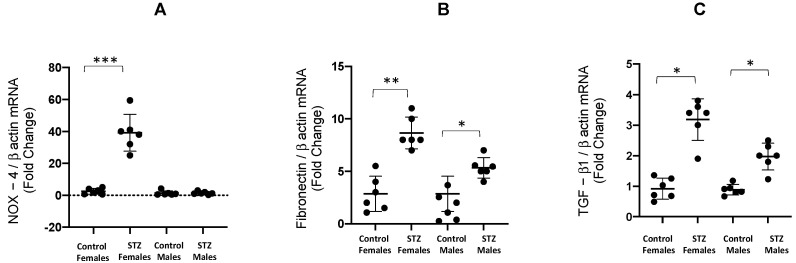
mRNA expression of NOX4, fibronectin and TGF-β1 in kidney medullary tissues from male and female control and STZ mice. After 6 days of single STZ administration, NOX-4 mRNA/β-actin mRNA ratio was significantly augmented in female STZ mice compared to control female mice. This increase was not observed in male STZ mice (**A**). Fibronectin mRNA/β-actin mRNA ratios were augmented in male STZ mice and, to a lesser extent, in females (**B**). TGF-β1 mRNA/β-actin mRNA ratios were augmented in STZ females and male mice (**C**). * *p* < 0.05 vs. control group, ** *p* < 0.01 vs. control group; *** *p* < 0.001 vs. control group.

**Figure 4 antioxidants-12-00729-f004:**
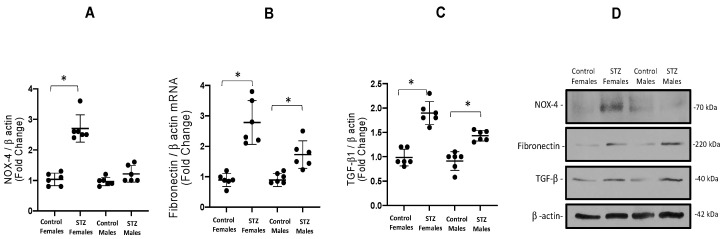
Protein expression of NOX-4, fibronectin and TGF-β1. (**A**). NOX-4 was augmented in STZ females, while males had no significant change. Protein abundances of fibronectin was augmented in both males and females (**B**). Similarly, TGF-β1 was also augmented in both STZ male and female mice (**C**). A representative blot from 4–6 experiments is shown in D. * *p* < 0.05 vs. control group.

**Figure 5 antioxidants-12-00729-f005:**
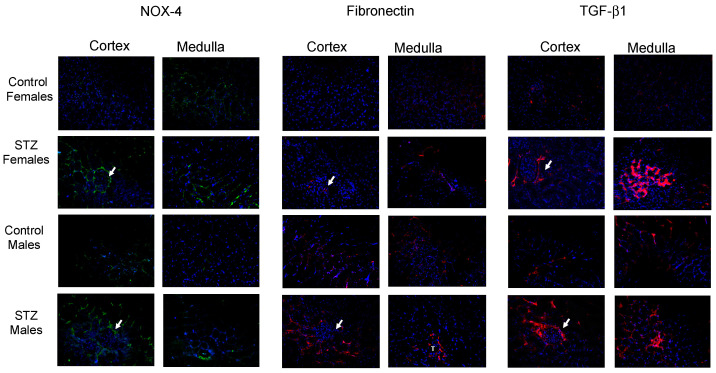
Immunohistochemistry in kidney medullary tissues from control and STZ female and male mice. Arrows indicate glomeruli. After 6 days of single STZ administration, NOX-4 (green color) was significantly augmented in cortical (glomeruli and tubular structures) and medullary tissues (tubular and interstitial structures) from female STZ mice. This was less evident in males. Fibronectin immunostaining was more evident in STZ mice, while TGF-β1 was more evident in glomeruli and tubular cells in female and male STZ mice compared to controls.

**Figure 6 antioxidants-12-00729-f006:**
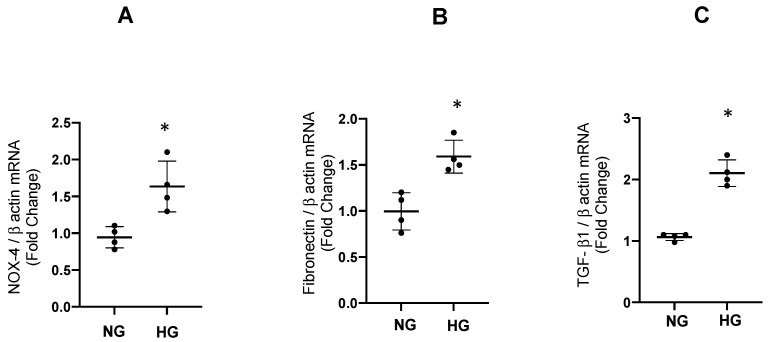
Expression of mRNA of NOX-4, fibronectin and TGF-β1. The abundance of mRNA of NOX-4, fibronectin, and TGF-β1 was augmented after 48 h of high glucose conditions (HG) as compared to normal glucose (NG) in IMCD cells. * *p* < 0.05 vs. control group, *n* = 4.

**Figure 7 antioxidants-12-00729-f007:**
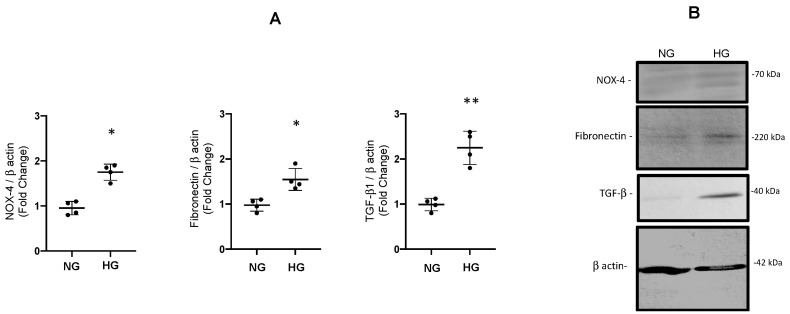
(**A**). Protein abundances of fibronectin, TGF-β1 1 and NOX-4 after 48 h of high glucose conditions (HG) vs. normal glucose (NG) assessed by immunoblot in IMCD cells. * *p* < 0.05 vs. control group, ** *p* < 0.01 vs. control group *n* = 4. (**B**). Representative blot for each protein analyzed. The β-actin was used as loading control.

**Figure 8 antioxidants-12-00729-f008:**
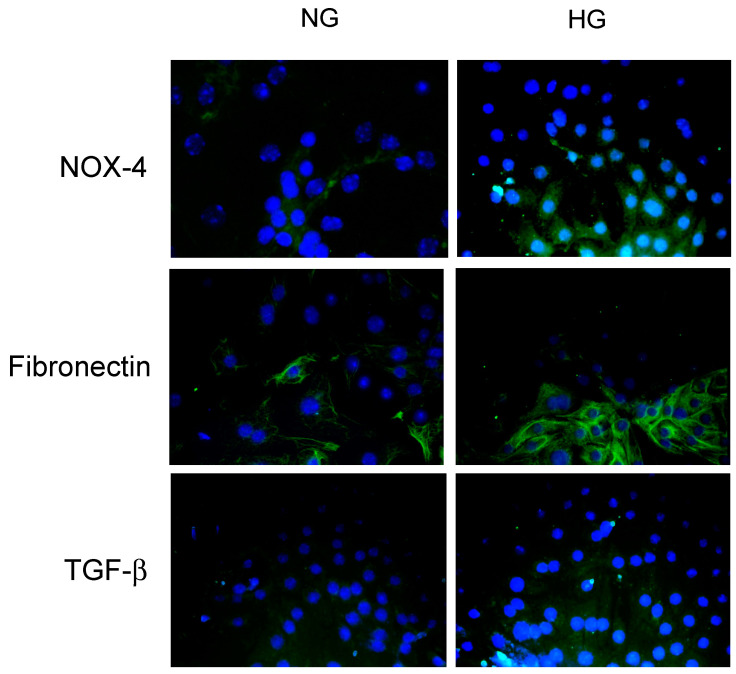
Immunofluorescence experiments in primary cultures of inner medullary collecting duct cells (IMCD). Fibronectin was augmented after 48 h of high glucose conditions (HG, 25 mM). The TGF-β1 antibody used in this study was not able to determine a specific labeling in cultured IMCD. NOX-4 labeling intensity was augmented after 48 h of HG.

**Figure 9 antioxidants-12-00729-f009:**
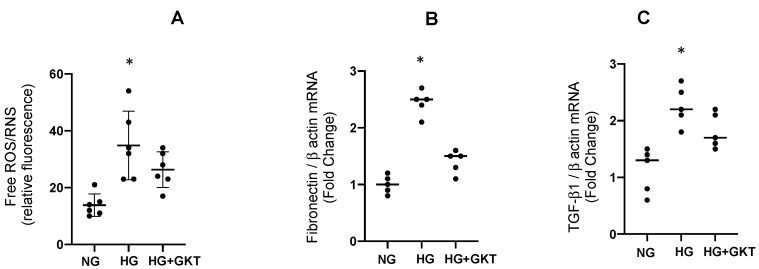
ROS, mRNA expression of fibronectin and TGF-β1 in mouse M1 collecting duct cell line treated with high glucose (HG) with or without GKT 137831 (GKT) treatment. (**A**) Effect of NOX-4 inhibition on ROS generation and transcript levels of fibronectin (**B**) and TGF-β1 (**C**) in M-1 collecting duct cell line. GKT 137831 partially blunted ROS induction and increased mRNA levels of fibronectin and TGF-β1. * *p* < 0.05, *n* = 6.

**Figure 10 antioxidants-12-00729-f010:**
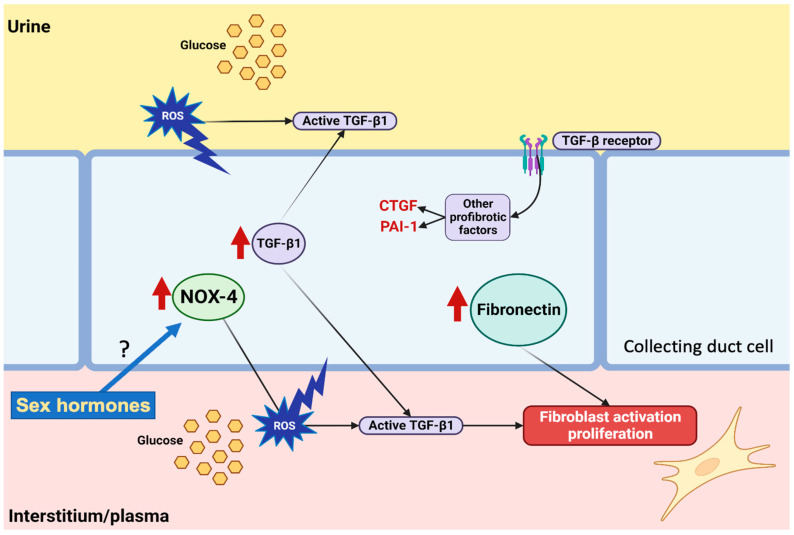
Hypothetical pathways involved in the regulation of reactive oxygen species (ROS) production under high glucose conditions. Hyperglycemia and glucosuria promotes auto-oxidation of glucose to form free radicals in collecting duct cells. In vivo data showed that NOX-4 is upregulated in females but not in males. TGF-β1 and fibronectin were increased in vivo and in vitro in collecting duct cells. Experiments on M-1 collecting duct cell line demonstrated that NOX-4 contributes to ROS production under high glucose conditions. Activation of the TGF-β1 receptor further promotes induction of other profibrotic factors such as CTGF and PAI-1, while fibronectin enhances fibroblast activation and fibroblast proliferation. Figure created with BioRender.com (BioRender’s Academic License BJ24Y7RW2L).

**Table 1 antioxidants-12-00729-t001:** Physiological parameters in male and female mice after 6 days of a single dose of streptozotocin. * *p* < 0.05, ** *p* < 0.01.

	Control Females	STZ Females	Control Males	STZ Males
Hct (%)	46 ± 6	48 ± 4	52 ± 5	48 ± 5
plasma Na^+^ (mEq/L)	147 ± 3	147 ± 4	145 ± 2	143 ± 4
plasma K^+^ (mEq/L)	3.3 ± 0.6	3.5 ± 0.3	3.4 ± 0.3	3.5 ± 0.3
plasma Cl^−^ (mEq/L)	106 ± 5	107 ± 3	106 ± 7	108 ± 5
Fasting blood glucose (mg/dL)	98 ± 25	368 ± 60 **	102 ± 32	385 ± 35 **
Serum creatinine (mg/dL)	0.22 ± 0.02	0.23 ± 0.04	0.21 ± 0.02	0.24 ± 0.05
Body weight (g)	35 ± 5	28 ± 3 *	34 ± 4	24 ± 5 *
Food intake (g)	2.5 ± 0.1	3.1 ± 0.2 *	2.3 ± 0.2	3.4 ± 0.3 *
Water intake (mL)	5.1 ± 0.2	9.1 ± 0.8 *	4.9 ± 0.7	12.4 ± 0.7 *
Urine output (mL)	2.2 ± 0.1	6.1 ± 0.7 *	2.6 ± 0.4 *	5.7 ± 0.6 *
Systolic blood pressure (mm Hg)	122 ± 5	129 ± 8	126 ± 5	125 ± 7

## Data Availability

The data presented in this study are available on request from the corresponding author.

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
