# Peer review of "Increased Renal Medullary NOX-4 in Female but Not Male Mice during the Early Phase of Type 1 Diabetes: Potential Role of ROS in Upregulation of TGF-β1 and Fibronectin in Collecting Duct Cells"

_antioxidants, 2023, doi:10.3390/antiox12030729_

Round 1

Reviewer 1 Report

I would like to congratulate the authors for interesting manuscript. Being said that I have following questions for the authors. 

1. in page 2, line 59, What does authors mean by pathogenetic influence? do they mean physiological abnormalities/changes? please clarify   

2. in page 2, third paragraph, authors need to cite recent publications to support authors statements/climes. 

3. in same paragraph, authors write "The exposure to high glucose levels promotes the 60 actions of injurious cytokines such as TGF-β1, as well as extracellular matrix proteins ". in this statement, was authors trying to say elivaed TGF and ECM levels? explain 

4. line 73, rewrite the sentence "literature are mainly focused on males " to "literature are mainly focused on male mice".

5. line 76-77, Rewrite the sentence in the past tense. 

6.  page 3, in blood glucose measurement study, when did authors gave glucose overload to mice? I mean after ATZ injection and is this glucose overload was given once or multiple times? also remove the sentence in line 108 as authors repeating the above sentence. 

7. rewrite the title of 2.3 to "sodium measurement from plasma and urine".

8. creatinine experiment, check the RPM levels if correct ? is it 10.000g or 10,000g

9. for western, where are the proteins derived from? Was the proteins derived from renal medulla or some other sources?

10. how did authors analyzed/imaged the blots? by IR or chemi?

11. for kidney immunofluorescence, How kidney slices were made? can authors include the brief procedure?

12. References are not in journal format. make necessary changes. 

Author Response

Author's Reply to the Review Report (Reviewer 1)

I would like to congratulate the authors for interesting manuscript. Being said that I have following questions for the authors. 

ANSWER: We appreciate the detailed critiques and the thought-provoking issues that have been raised. We have carefully considered all the comments and revised the paper point-by-point. Our detailed responses are as follows (modifications were made in red in the new version of the manuscript).

  1. in page 2, line 59, What does authors mean by pathogenetic influence? do they mean physiological abnormalities/changes? please clarify 

ANSWER: We appreciate the reviewer’s comment. In the revised manuscript, we have modified the statement to make it clearer and easier to understand  . “…. Because the renal medullary tissues and in particular the collecting ducts are more likely to be damaged when limited glucose transport in the proximal tubule has been exceeded, their resorptive function is especially affected by the pathological mechanisms activated by hyperglycemia and hyperglycosuria….”

  1. in page 2, third paragraph, authors need to cite recent publications to support authors statements/climes. 

ANSWER: According to the reviewer’s comment, , in the revised manuscript, we have included 4 new references in paragraph 3 to support our statements.

  1. in same paragraph, authors write "The exposure to high glucose levels promotes the 60 actions of injurious cytokines such as TGF-β1, as well as extracellular matrix proteins ". in this statement, was authors trying to say elivaed TGF and ECM levels? explain 

ANSWER: we apologize if our sentence was confusing. In the revised manuscript we have reworded the paragraph to make it clearer and easier to understand ….”The exposure to high glucose levels promotes an increase in cytokines levels such as TGF-β1, as well as extracellular matrix formation in the kidney”….

  1. line 73, rewrite the sentence "literature are mainly focused on males " to "literature are mainly focused on male mice".

ANSWER: Thank you for your comment we have reworded the sentence accordingly in the revised manuscript

  1. line 76-77, Rewrite the sentence in the past tense. 

ANSWER: Thank you for your comment we have reworded the sentence accordingly in the revised manuscript.

  1.  page 3, in blood glucose measurement study, when did authors gave glucose overload to mice? I mean after ATZ injection and is this glucose overload was given once or multiple times? also remove the sentence in line 108 as authors repeating the above sentence. 

ANSWER: Thank you for your comment. We have changed the sentence in the revised manuscript ….”the glucose overload test was performed once at day 5 after STZ treatment. The blood glucose was measured before and after 15, 30, 60 and 120 min of a glucose overload of 2 g/kg BW, i.p. Blood glucose was measured every 15 min”….

  1. rewrite the title of 2.3 to "sodium measurement from plasma and urine".

ANSWER: Thank you for your comment we have reworded the sentence accordingly in the revised manuscript

  1. creatinine experiment, check the RPM levels if correct? is it 10.000g or 10,000g

ANSWER: Thank you for your comment we have reworded the sentence in the revised manuscript: … “Blood was collected from the cardiac puncture and centrifuged at 10,000 g for 5 min at 4°C to collect the plasma”….

  1. for western, where are the proteins derived from? Was the proteins derived from renal medulla or some other sources?

ANSWER: Thank you for your suggestion, we have included a new paragraph in the revised manuscript. Please refer to the method section. -…”Inner sections of kidney poles were minced and homogenized in ice-cold RIPA buffer with protease inhibitors (0.15 M NaCl, 50 mM Tris-HCl pH 8.0, 1% NP-40, 0.5% deoxycholate). Total protein content was determined by the BCA Protein Assay Kit (Santa Cruz) according to manufacturer instructions”….  

  1. how did authors analyzed/imaged the blots? by IR or chemi?

ANSWER: Thank you for your comment. We added the following sentence in the new version of the manuscript: …:”Blots were incubated with chemiluminescent substrate ECL reagent (Perkin Elmer) by direct exposure on films that were scanned and analyzed by ImageJ software”.

  1. for kidney immunofluorescence, How kidney slices were made? can authors include the brief procedure?

ANSWER: Thank you for your comment. We have included a new sentence in the revised manuscript: …”For histological studies, a kidney pole will be fixed with Bouin's solution (picric acid, formaldehyde and acetic acid 4%) overnight and then subjected to inclusion into paraffin blocks. Kidney poles were cut in 3μm sections using microtome (HM325, Thermo)”…

  1. References are not in journal format. make necessary changes. 

ANSWER: We appreciate your comment, we have downloaded the correct format. Thank you.

Reviewer 2 Report

antioxidants-2267636

Increased renal medullary NOX-4 in female but not male mice during the early phase of type 1 diabetes: potential role of ROS in upregulation of TGF-β1 and Fibronectin in collecting duct cells.

Felipe Casado-Barragán , Geraldine Lazcano-Páez , Paulina E Larenas , Monserrat Aguirre-Delgadillo , Fernanda Olivares-Aravena , Daniela Witto-Oyarce , Camila Núñez-Allimant , Katherin Silva , Quynh My Nguyen , Pilar Cárdenas , Modar Kassan , Alexis A Gonzalez *

In this study authors evaluated the role of ROS production in altering the expression of fibrotic markers in collecting duct cells type 1 diabetes. They found that TGF-beta1, fibronectin or NOX-4 were upregulated with differences either in gender or in glucose conditions.

In this work the potential mechanisms of tubular damage associated to ROS production was explored and an

hypothesis upon results was formulated.

Major and minor comments

The materials and methods section should be better described. I would suggest preparing a graphical abstract showing all methods in your study.

The authors must indicate how mice were sacrificed, conservation of tissue samples and how these specimens were processed for further evaluations.

Which samples underwent to western blotting analysis?

In figure 6 bans in blots, especially for fibronectin and NOX-4, are not very visible.

In results, lane 197, please specify the abbreviation.

In the discussion, the authors should mention in discussion the limitation of the study.

Author Response

Author's Reply to the Review Report (Reviewer 2)

In this study authors evaluated the role of ROS production in altering the expression of fibrotic markers in collecting duct cells type 1 diabetes. They found that TGF-beta1, fibronectin or NOX-4 were upregulated with differences either in gender or in glucose conditions.

In this work the potential mechanisms of tubular damage associated to ROS production was explored and an hypothesis upon results was formulated. 

ANSWER: We appreciate your inputs. We have carefully considered all the comments and revised the paper point-by-point. Our detailed responses are as follows (modifications were made in red in the new version of the manuscript).

Major and minor comments

  1. The materials and methods section should be better described. I would suggest preparing a graphical abstract showing all methods in your study.

ANSWER: Thank you for your comment, in the revised manuscript we have included a new Figure 1 showing all methods in this study.

  1. The authors must indicate how mice were sacrificed, conservation of tissue samples and how these specimens were processed for further evaluations.

ANSWER: Thank you for your comment. Please see the new version of the manuscript where we better described all the methods used (yellow highlighted).

  1. Which samples underwent to western blotting analysis?

ANSWER: Inner medullary tissues from left kidney poles were extracted for protein analysis and (2 mm x 2 mm) and 1mm x 1mm piece used for total ARN extractions. Right kidney was used for immunostainings (kidney poles including medullary areas) and ROS measurements. Please see the new version of the manuscript (yellow highlighted) for a better comprehension of the methodology.

  1. In figure 6 bans in blots, especially for fibronectin and NOX-4, are not very visible.

ANSWER: Thank you for your suggestions. We have included a new image in the new version of the manuscript, according to the requirements of the journal.

  1. In results, lane 197, please specify the abbreviation.

ANSWER: Thank you for your comment, abbreviation was specified in the revised manuscript.

  1. In the discussion, the authors should mention in discussion the limitation of the study.

ANSWER: Thank you for your comment, in the revised manuscript, we added a new paragraph indicating the limitations of this study.

Round 2

Reviewer 1 Report

I would like to thank the editor for sending revised manuscript. Authors added all the requested modifications to the manuscript and it look much better in its Current form.   

Reviewer 2 Report

Authors addressed all the reviewer’s comments improving the manuscript as suggested. There is no further comment, and, in my opinion, the paper is suitable for publication.